# Learning discrete distributions with infinite support

**Doron Cohen**
Department of Computer Science
Ben-Gurion University of the Negev
Beer-Sheva, Israel
doronv@post.bgu.ac.il

**Aryeh Kontorovich**
Department of Computer Science
Ben-Gurion University of the Negev
Beer-Sheva, Israel
karyeh@cs.bgu.ac.il

**Geoffrey Wolfer**
Department of Computer Science
Ben-Gurion University of the Negev
Beer-Sheva, Israel
geoffrey@post.bgu.ac.il

## Abstract

We present a novel approach to estimating discrete distributions with (potentially) infinite support in the total variation metric. In a departure from the established paradigm, we make no structural assumptions whatsoever on the sampling distribution. In such a setting, distribution-free risk bounds are impossible, and the best one could hope for is a fully empirical data-dependent bound. We derive precisely such bounds, and demonstrate that these are, in a well-defined sense, the best possible. Our main discovery is that the half-norm of the empirical distribution provides tight upper and lower estimates on the empirical risk. Furthermore, this quantity decays at a nearly optimal rate as a function of the true distribution. The optimality follows from a minimax result, of possible independent interest. Additional structural results are provided, including an exact Rademacher complexity calculation and apparently a first connection between the total variation risk and the missing mass.

## 1 Introduction

Estimating a discrete distribution in the total variation (TV) metric is a central problem in computer science and statistics (see, e.g., Han et al. [2015], Kamath et al. [2015], Orlitsky and Suresh [2015] and the references therein). The TV metric, which we use throughout the paper, is a natural and abundantly motivated choice [Devroye and Lugosi, 2001]. For support size $d$, a sample of size $\mathcal{O}(d/\varepsilon^2)$ suffices for the maximum-likelihood estimator (MLE) to be $\varepsilon$-close (with constant probability) to the unknown target distribution. A matching lower bound is known [Anthony and Bartlett, 1999], and has been computed down to the exact constants [Kamath et al., 2015].

Classic VC theory — and, in particular, the aforementioned results — imply that for infinite support, no distribution-free sample complexity bound is possible. If $\boldsymbol{\mu}$ is the target distribution and $\widehat{\boldsymbol{\mu}}_m$ is its empirical (i.e., MLE) estimate based on $m$ iid samples, then Berend and Kontorovich [2013] showed that

$$\frac{1}{4}\Lambda_m(\boldsymbol{\mu}) - \frac{1}{4\sqrt{m}} \;\leq\; \mathbb{E}\left[\|\boldsymbol{\mu} - \widehat{\boldsymbol{\mu}}_m\|_{\mathsf{TV}}\right] \;\leq\; \Lambda_m(\boldsymbol{\mu}), \qquad m \geq 2, \tag{1}$$

where

$$\Lambda_m(\boldsymbol{\mu}) = \sum_{j\in\mathbb{N}:\boldsymbol{\mu}(j)<1/m} \boldsymbol{\mu}(j) + \frac{1}{2\sqrt{m}} \sum_{j\in\mathbb{N}:\boldsymbol{\mu}(j)\geq 1/m} \sqrt{\boldsymbol{\mu}(j)}. \tag{2}$$

The quantity $\Lambda_m(\boldsymbol{\mu})$ has the advantage of always being finite and of decaying to 0 as $m \to \infty$. The bound in (1) suggests that $\Lambda_m(\boldsymbol{\mu})$, or a closely related measure, controls the sample complexity for learning discrete distributions in TV. Further supporting the foregoing intuition is the observation that for finite support size $d$ and $m \gg 1$, we have $\Lambda_m \lesssim \sqrt{d/m}$, recovering the known minimax rate. Additionally, a closely related measure turns out to control a minimax risk rate in a sense made precise in Theorem 2.5.

One shortcoming of (1) is that the lower bound only holds for the MLE, leaving the possibility that a different estimator could achieve significantly improved bounds. Another shortcoming of (1) and related estimates is that they are not *empirical*, in that they depend on the unknown quantity we are trying to estimate. A fully empirical bound, on the other hand, would give a high-probability estimate on $\|\boldsymbol{\mu} - \widehat{\boldsymbol{\mu}}_m\|_{\mathsf{TV}}$ solely in terms of observable quantities such as $\widehat{\boldsymbol{\mu}}_m$. Of course, such a bound should also be non-trivial, in the sense of improving with growing sample size and approaching 0 as $m \to \infty$. A further desideratum might be something akin to *instance optimality*: We would like the rate at which the empirical bound decays to be "the best" possible for the given $\boldsymbol{\mu}$, in an appropriate sense. Our analogue of instance optimality is inspired by, but distinct from, that of Valiant and Valiant [2016], as discussed in detail in Related work below.

**Our contributions.** We address the shortcomings of existing estimators detailed above by providing a fully empirical bound on $\|\boldsymbol{\mu} - \widehat{\boldsymbol{\mu}}_m\|_{\mathsf{TV}}$. Our main discovery is that the quantity $\Phi_m(\widehat{\boldsymbol{\mu}}_m) := \frac{1}{\sqrt{m}} \sum_{j \in \mathbb{N}} \sqrt{\widehat{\boldsymbol{\mu}}_m(j)}$ satisfies all of the desiderata posed above for an empirical bound. As we show in Theorems 2.1 and 2.2, $\Phi_m(\widehat{\boldsymbol{\mu}}_m)$ provides tight, high-probability upper and lower bounds on $\|\boldsymbol{\mu} - \widehat{\boldsymbol{\mu}}_m\|_{\mathsf{TV}}$. Further, Theorem 2.3 shows that $\mathbb{E}\left[\Phi_m(\widehat{\boldsymbol{\mu}}_m)\right]$ behaves as $\Lambda_m(\boldsymbol{\mu})$ defined in (2). Finally, a result in the spirit of instance optimality, Theorem 2.4, shows that no other estimator-bound pair can improve upon $(\widehat{\boldsymbol{\mu}}_m, \Phi_m)$, other than by small constants. The latter follows from a minimax bound of independent interest, Theorem 2.5. Additional structural results are provided, including an exact Rademacher complexity calculation and a connection (apparently the first) between the total variation risk and the missing mass.

**Definitions, notation and setting.** As we are dealing with discrete distributions, there is no loss of generality in taking our sample space to be the natural numbers $\mathbb{N} = \{1, 2, 3, \ldots\}$. For $k \in \mathbb{N}$, we write $[k] := \{i \in \mathbb{N} : i \le k\}$. The set of all distributions on $\mathbb{N}$ will be denoted by $\Delta_{\mathbb{N}}$, which we enlarge to include the "deficient" distributions: $\Delta_{\mathbb{N}} \subset \Delta_{\mathbb{N}}^{\circ} := \left\{\boldsymbol{\mu} \in [0,1]^{\mathbb{N}} : \sum_{i \in \mathbb{N}} \boldsymbol{\mu}(i) \le 1\right\}$. For $d \in \mathbb{N}$, we write $\Delta_d \subset \Delta_{\mathbb{N}}$ to denote those $\boldsymbol{\mu}$ whose support is contained in $[d]$.

For $\boldsymbol{\mu} \in \Delta_{\mathbb{N}}^{\circ}$ and $I \subseteq \mathbb{N}$, we write $\boldsymbol{\mu}(I) = \sum_{i \in I} \boldsymbol{\mu}(i)$. We define the *decreasing permutation* of $\boldsymbol{\mu} \in \Delta_{\mathbb{N}}^{\circ}$, denoted by $\boldsymbol{\mu}^{\downarrow}$, to be the sequence $(\boldsymbol{\mu}(i))_{i \in \mathbb{N}}$ sorted in non-increasing order, achieved by a[1] permutation $\Pi_{\boldsymbol{\mu}}^{\downarrow} : \mathbb{N} \to \mathbb{N}$; thus, $\boldsymbol{\mu}^{\downarrow}(i) = \boldsymbol{\mu}(\Pi_{\boldsymbol{\mu}}^{\downarrow}(i))$. For $0 < \eta < 1$, define $T_{\boldsymbol{\mu}}(\eta) \in \mathbb{N}$ as the least $t$ for which $\sum_{i>t}^{\infty} \boldsymbol{\mu}^{\downarrow}(i) < \eta$. This induces a truncation of $\boldsymbol{\mu}$, denoted by $\boldsymbol{\mu}[\eta] \in \Delta_{\mathbb{N}}^{\circ}$ and defined by $\boldsymbol{\mu}[\eta](i) = \mathbf{1}[\Pi_{\boldsymbol{\mu}}^{\downarrow}(i) \le T_{\boldsymbol{\mu}}(\eta)]\boldsymbol{\mu}(i)$.

For $\boldsymbol{\mu}, \boldsymbol{\nu} \in \Delta_{\mathbb{N}}^{\circ}$, we define the *total variation distance* in terms of the $\ell_1$ norm:

$$\|\boldsymbol{\mu} - \boldsymbol{\nu}\|_{\mathsf{TV}} := \frac{1}{2}\|\boldsymbol{\mu} - \boldsymbol{\nu}\|_1 = \frac{1}{2}\sum_{i \in \mathbb{N}} |\boldsymbol{\mu}(i) - \boldsymbol{\nu}(i)|. \tag{3}$$

For $\boldsymbol{\mu} \in \Delta_{\mathbb{N}}^{\circ}$, we also define the *half-norm*[2] as

$$\|\boldsymbol{\mu}\|_{1/2} := \left(\sum_{i \in \Omega} \sqrt{\boldsymbol{\mu}(i)}\right)^2; \tag{4}$$

note that while $\|\boldsymbol{\mu}\|_{1/2}$ may be infinite, we have $\|\boldsymbol{\mu}\|_{1/2} \le \|\boldsymbol{\mu}\|_0$, where the latter denotes the support size.

For $m \in \mathbb{N}$ and $\boldsymbol{\mu} \in \Delta_{\mathbb{N}}$, we write $\boldsymbol{X} = (X_1, \ldots, X_m) \sim \boldsymbol{\mu}^m$ to mean that the components of the vector $\boldsymbol{X}$ are drawn iid from from $\boldsymbol{\mu}$. We reserve $\widehat{\boldsymbol{\mu}}_m \in \Delta_{\mathbb{N}}$ for the empirical measure induced by the sample $\boldsymbol{X}$, i.e. $\widehat{\boldsymbol{\mu}}_m(i) := \frac{1}{m}\sum_{t \in [m]} \mathbf{1}[X_t = i]$; the term MLE will be used interchangeably.

For the class of boolean functions over the integers $\{f\colon \mathbb{N} \to \{0,1\}\}$, which we denote by $\{0,1\}^{\mathbb{N}}$, recall the definition of the *empirical Rademacher complexity* [Mohri et al., 2012, Definition 3.1] conditional on the sample $\boldsymbol{X}$:

$$\hat{\mathfrak{R}}_m(\boldsymbol{X}) := \mathbb{E}_{\boldsymbol{\sigma}}\left[ \sup_{f \in \{0,1\}^{\mathbb{N}}} \frac{1}{m} \sum_{t=1}^{m} \sigma_t f(X_t) \right], \tag{5}$$

where $\boldsymbol{\sigma} = (\sigma_1, \dots, \sigma_m) \sim \mathrm{Uniform}(\{-1,1\}^m)$. The expectation of the above random quantity is the *Rademacher complexity* [Mohri et al., 2012, Definition 3.2]:

$$\mathfrak{R}_m := \mathbb{E}_{\boldsymbol{X} \sim \boldsymbol{\mu}^m}\left[ \hat{\mathfrak{R}}(\boldsymbol{X}) \right]. \tag{6}$$

**Related work.** Given the classical nature of the problem, a comprehensive literature survey is beyond our scope; the standard texts Devroye and Györfi [1985], Devroye and Lugosi [2001] provide much of the requisite background. Chapter 6.5 of the latter makes a compelling case for the TV metric used in this paper, but see Waggoner [2015] and the works cited therein for results on other $\ell_p$ norms. Though surveying all of the relevant literature is a formidable task, a relatively streamlined narrative may be distilled. Conceptually, the simplest case is that of $\|\boldsymbol{\mu}\|_0 < \infty$ (i.e., finite support). Since learning a distribution over $[d]$ in TV is equivalent to agnostically learning the function class $\{0,1\}^d$, standard VC theory [Anthony and Bartlett, 1999, Kontorovich and Pinelis, 2019] entails that the MLE achieves the minimax risk rate of $\sqrt{d/m}$ over all $\boldsymbol{\mu} \in \Delta_{\mathbb{N}}$ with $\|\boldsymbol{\mu}\|_0 \leq d$. An immediate consequence is that in order to obtain quantitative risk rates for the case of infinite support, one must assume some sort of structure [Diakonikolas, 2016]. One can, for example, obtain minimax rates for $\boldsymbol{\mu}$ with bounded entropy [Han et al., 2015], or, say, bounded half-norm (as we do here). Alternatively, one can restrict one's attention to a finite class $\mathcal{Q} \subset \Delta_{\mathbb{N}}$; here too, optimal results are known [Bousquet et al., 2019]. Berend and Kontorovich [2013] was one of the few works that made no assumptions on $\boldsymbol{\mu} \in \Delta_{\mathbb{N}}$, but only gave non-empirical bounds.

Our work departs from the paradigm of a-priori constraints on the unknown sampling distribution. Instead, our estimates hold for all $\boldsymbol{\mu} \in \Delta_{\mathbb{N}}$. Of course, this must come at a price: no a-priori sample complexity bounds are possible in this setting. Absent any prior knowledge regarding $\boldsymbol{\mu}$, one can only hope for sample-dependent *empirical* bounds, and we indeed obtain these. Further, our empirical bounds are essentially the best possible, as formalized in Theorem 2.4. The latter result may be thought of as a learning-theoretic analogue of being *instance-optimal*, as introduced by Valiant and Valiant [2017] in the testing framework. Instance optimality is a very natural notion in the context of testing whether an unknown sampling distribution $\boldsymbol{\mu}$ is identical to or $\varepsilon$-far from a given reference one, $\boldsymbol{\mu}_0$. For example, Valiant and Valiant discovered that a truncated $2/3$-norm of $\boldsymbol{\mu}_0$ — i.e., a quantity closely related to $\|\boldsymbol{\mu}_0\|_{2/3}$ — controls the complexity of the testing problem in TV distance. Instance optimality is more difficult to formalize for distribution learning, since for any given $\boldsymbol{\mu} \in \Delta_{\mathbb{N}}$, there is a trivial "learner" with $\boldsymbol{\mu}$ hard-coded inside. Valiant and Valiant [2016] defined this notion in terms of competing against an oracle who knows the distribution up to a permutation of the atoms, and did not provide empirical confidence intervals. We do derive fully empirical bounds, and further show that they are impossible to improve upon — by *any estimator* — other than by constants. Our results suggest that the half-norm $\|\boldsymbol{\mu}\|_{1/2}$ plays a role in learning analogous to that of $\|\boldsymbol{\mu}\|_{2/3}$ in testing. As an intriguing aside, we note that the half-norm corresponds to the Tsallis $q$-entropy with $q = 1/2$, which was shown to be an optimal regularizer in some stochastic and adversarial bandit settings [Zimmert and Seldin, 2019]. We leave the question of investigating a deeper connection between the two results for future work.

## 2 Main results

In this section, we formally state our main results. Recall from the Definitions that the sample $\boldsymbol{X} = (X_1, \dots, X_m) \sim \boldsymbol{\mu}^m$ induces the empirical measure (MLE) $\widehat{\boldsymbol{\mu}}_m$, and that a key quantity in our bounds is

$$\Phi_m(\widehat{\boldsymbol{\mu}}_m) \;=\; \frac{1}{\sqrt{m}}\|\widehat{\boldsymbol{\mu}}_m\|_{1/2}^{1/2} \;=\; \frac{1}{\sqrt{m}}\sum_{j \in \mathbb{N}} \sqrt{\widehat{\boldsymbol{\mu}}_m(j)}. \tag{7}$$

Our first result is a fully empirical, high-probability upper bound on $\|\widehat{\boldsymbol{\mu}}_m - \boldsymbol{\mu}\|_{\mathrm{TV}}$ in terms of $\Phi_m(\widehat{\boldsymbol{\mu}}_m)$:

**Theorem 2.1.** *For all $m \in \mathbb{N}$, $\delta \in (0, 1)$, and $\boldsymbol{\mu} \in \Delta_\mathbb{N}$, we have that*

$$\|\widehat{\boldsymbol{\mu}}_m - \boldsymbol{\mu}\|_{\mathsf{TV}} \leq \Phi_m(\widehat{\boldsymbol{\mu}}_m) + 3\sqrt{\frac{\log \frac{2}{\delta}}{2m}}$$

*holds with probability at least $1 - \delta$. We also have*

$$\mathbb{E}\left[\|\widehat{\boldsymbol{\mu}}_m - \boldsymbol{\mu}\|_{\mathsf{TV}}\right] \leq \mathbb{E}\left[\Phi_m(\widehat{\boldsymbol{\mu}}_m)\right].$$

Since $\|\widehat{\boldsymbol{\mu}}_m\|_{1/2} \leq \|\widehat{\boldsymbol{\mu}}_m\|_0 \leq \|\boldsymbol{\mu}\|_0$, this recovers the minimax rate of $\sqrt{d/m}$ for $\boldsymbol{\mu} \in \Delta_\mathbb{N}$ with $\|\boldsymbol{\mu}\|_0 \leq d$. We also provide a matching lower bound:

**Theorem 2.2.** *For all $m \in \mathbb{N}$, $\delta \in (0, 1)$, and $\boldsymbol{\mu} \in \Delta_\mathbb{N}$, we have that*

$$\|\widehat{\boldsymbol{\mu}}_m - \boldsymbol{\mu}\|_{\mathsf{TV}} \geq \frac{1}{4\sqrt{2}}\Phi_m(\widehat{\boldsymbol{\mu}}_m) - 3\sqrt{\frac{\log \frac{2}{\delta}}{m}}$$

*holds with probability at least $1 - \delta$.*

Our empirical measure $\Phi_m(\widehat{\boldsymbol{\mu}}_m)$ is never much worse than the non-empirical $\Lambda_m(\boldsymbol{\mu})$, defined in (2):

**Theorem 2.3.** *For all $m \in \mathbb{N}$ and $\boldsymbol{\mu} \in \Delta_\mathbb{N}$ we have*

$$\mathbb{E}\left[\Phi_m(\widehat{\boldsymbol{\mu}}_m)\right] \leq 2\Lambda_m(\boldsymbol{\mu})$$

*and, with probability at least $1 - \delta$,*

$$\Phi_m(\widehat{\boldsymbol{\mu}}_m) \leq 2\Lambda_m(\boldsymbol{\mu}) + \sqrt{\log(1/\delta)/m}.$$

Furthermore, no other estimator-bound pair $(\tilde{\boldsymbol{\mu}}_m, \Psi_m)$ can improve upon $(\widehat{\boldsymbol{\mu}}_m, \Phi_m)$, other than by a constant. This is the "instance optimality" result alluded to above:

**Theorem 2.4.** *There exist universal constants $a, b > 0$ such that the following holds. For any estimator-bound pair $(\tilde{\boldsymbol{\mu}}_m, \Psi_m)$ and any* continuous *function $\theta \colon \mathbb{R}_+ \to \mathbb{R}_+$ such that*

$$\mathbb{E}\left[\|\tilde{\boldsymbol{\mu}}_m - \boldsymbol{\mu}\|_{\mathsf{TV}}\right] \leq \mathbb{E}\left[\Psi_m(\tilde{\boldsymbol{\mu}}_m)\right] \leq \theta\left(\mathbb{E}\left[\Phi_m(\widehat{\boldsymbol{\mu}}_m)\right]\right)$$

*holds for all $\boldsymbol{\mu} \in \Delta_\mathbb{N}$, $\theta$ necessarily verifies*

$$\inf_{0 < x < b} \frac{\theta(x)}{x} \geq \frac{1}{a}.$$

The next result, framed in the high-probability setting, draws a direct parallel between our characterization of the learning sample complexity via the half-norm and Valiant and Valiant [2017]'s characterization of the testing sample complexity via the 2/3-norm. The truncation is needed to ensure finiteness, since the $\|\boldsymbol{\mu}\|_{1/2} = \infty$ for heavy-tailed distributions (e.g. $\boldsymbol{\mu}(i) \propto 1/i^2$).

**Theorem 2.5.** *There is a universal constant $C > 0$ such that for all $\Lambda \geq 2$ and $0 < \varepsilon, \delta < 1$, the MLE $\widehat{\boldsymbol{\mu}}_m$ verifies the following optimality property: For all $\boldsymbol{\mu} \in \Delta_\mathbb{N}$ with $\|\boldsymbol{\mu}[2\varepsilon\delta/9]\|_{1/2} \leq \Lambda$, we have*

$$m \geq C\varepsilon^{-2} \max\{\Lambda, \log(1/\delta)\} \implies \mathbb{P}\left(\|\widehat{\boldsymbol{\mu}}_m - \boldsymbol{\mu}\|_{\mathsf{TV}} < \varepsilon\right) \geq 1 - \delta.$$

*On the other hand, for* any *estimator $\tilde{\boldsymbol{\mu}}_m \colon \mathbb{N}^m \to \Delta_\mathbb{N}$ there is a $\boldsymbol{\mu} \in \Delta_\mathbb{N}$ with $\max\left\{\|\boldsymbol{\mu}[\varepsilon/18]\|_{1/2}, \|\boldsymbol{\mu}[2\varepsilon\delta/9]\|_{1/2}\right\} \leq \Lambda$ such that:*

$$m < C\varepsilon^{-2} \min\{\Lambda, \log(1/\delta)\} \implies \mathbb{P}\left(\|\tilde{\boldsymbol{\mu}}_m - \boldsymbol{\mu}\|_{\mathsf{TV}} \geq \varepsilon\right) \geq \min\{3/4, 1 - \delta\}.$$

The above is a simplified statement chosen for brevity; a considerably refined version is stated and proved in Theorem 3.1.

# 3 Proofs

## 3.1 Proof of Theorem 2.1

The proof consists of two parts. The first is contained in Lemma 3.1, which provides a high-probability empirical upper bound, and an expectation bound, similar to Theorem 2.1, but in terms of $\hat{\mathfrak{R}}_m(\boldsymbol{X})$ instead of $\Phi_m(\widehat{\boldsymbol{\mu}}_m)$. The second part, contained in Lemma 3.2, provides an estimate of $\hat{\mathfrak{R}}_m(\boldsymbol{X})$ in terms of $\Phi_m(\widehat{\boldsymbol{\mu}}_m)$.

**Lemma 3.1.** *For all $m \in \mathbb{N}$, $\delta \in (0,1)$, and $\boldsymbol{\mu} \in \Delta_{\mathbb{N}}$, we have that*

$$\|\widehat{\boldsymbol{\mu}}_m - \boldsymbol{\mu}\|_{\mathsf{TV}} \leq 2\hat{\mathfrak{R}}_m(\boldsymbol{X}) + 3\sqrt{\frac{\log \frac{2}{\delta}}{2m}}$$

*holds with probability at least $1 - \delta$. We also have,*

$$\mathbb{E}\left[\|\widehat{\boldsymbol{\mu}}_m - \boldsymbol{\mu}\|_{\mathsf{TV}}\right] \leq 2\mathfrak{R}_m. \tag{8}$$

*Proof.* The high-probability bound from the observation,

$$\|\widehat{\boldsymbol{\mu}}_m - \boldsymbol{\mu}\|_{\mathsf{TV}} := \sup_{A \subseteq \mathbb{N}} \left(\boldsymbol{\mu}(A) - \widehat{\boldsymbol{\mu}}_m(A)\right) = \sup_{f \in \mathcal{F}} \left(\mathbb{E}_{X \sim \boldsymbol{\mu}}[f(X)] - \frac{1}{m}\sum_{i=1}^{m} f(X_i)\right) \tag{9}$$

where $\mathcal{F} := \{\mathbb{I}_A | A \subseteq \mathbb{N}\} = \{0,1\}^{\mathbb{N}}$, combined with [Mohri et al., 2012, Theorem 3.3], which states: Let $\mathcal{G}$ be a family of functions from $\mathcal{Z}$ to $[0,1]$ and let $\boldsymbol{\nu}$ be a distribution supported on a subset of $\mathcal{Z}$. Then, for any $\delta > 0$, with probability at least $1 - \delta$ over $\boldsymbol{Z} = (Z_1, \ldots, Z_m) \sim \boldsymbol{\nu}^m$, the following holds:

$$\sup_{g \in \mathcal{G}} \left(\mathbb{E}_{Z \sim \boldsymbol{\nu}}[g(Z)] - \frac{1}{m}\sum_{i=1}^{m} g(Z_i)\right) \leq 2\hat{\mathfrak{R}}_m(\boldsymbol{Z}) + 3\sqrt{\frac{\log \frac{2}{\delta}}{2m}}.$$

Plugging in $\mathcal{F}$ for $\mathcal{G}$ and $\boldsymbol{\mu}$ for $\boldsymbol{\nu}$ in the above theorem completes the proof of the high-probability bound. The expectation bound (eq. (8)) follows from the observation at eq. (9) and a symmetrization argument [Mohri et al., 2012, eq. (3.8) to (3.13)]. $\square$

In order to complete the proof, we apply

**Lemma 3.2** (Empirical Rademacher estimates). *Let $\boldsymbol{X} = (X_1, \ldots, X_m)$ and let $\widehat{\boldsymbol{\mu}}_m$ be the empirical measure constructed from the sample $\boldsymbol{X}$. Then,*

$$\frac{1}{2\sqrt{2}}\Phi_m(\widehat{\boldsymbol{\mu}}_m) \leq \hat{\mathfrak{R}}_m(\boldsymbol{X}) \leq \frac{1}{2}\Phi_m(\widehat{\boldsymbol{\mu}}_m).$$

*Proof.* The proof is based on an argument that was also developed in [Scott and Nowak, 2006, Section 7.1, Appendix E.] in the context of histograms and dyadic decision trees, and that was credited to Gilles Blanchard.

Let $\hat{S} = \{X_i | i \in [m]\}$ be the empirical support according to the sample $\boldsymbol{X} = (X_1, X_2, ..., X_m)$. Then,

$$m\hat{\mathfrak{R}}_m(\boldsymbol{X}) = \mathbb{E}_{\boldsymbol{\sigma}}\left[\sup_{f \in \{0,1\}^{\mathbb{N}}} \sum_{i=1}^{m} \sigma_i f(X_i)\right] = \mathbb{E}_{\boldsymbol{\sigma}}\left[\sup_{A \subseteq \hat{S}} \sum_{i=1}^{m} \sigma_i \mathbb{I}_A(X_i)\right]$$

$$= \sum_{x \in \hat{S}} \mathbb{E}_{\boldsymbol{\sigma}}\left[\sup_{A \subseteq \{x\}} \sum_{i:X_i=x} \sigma_i \mathbb{I}_A(X_i)\right] = \sum_{x \in \hat{S}} \mathbb{E}_{\boldsymbol{\sigma}}\left[\left(\sum_{i:X_i=x} \sigma_i\right)_+\right] = \sum_{x \in \hat{S}} \frac{1}{2} \mathbb{E}_{\boldsymbol{\sigma}}\left[\left|\sum_{i=1}^{m\widehat{\boldsymbol{\mu}}_m(x)} \sigma_i\right|\right],$$

where the last equality follows from counting $\{i : X_i = x\}$ and the symmetry of the random variable $\sum_{i=1}^{m} \sigma_i$ for all $n \in \mathbb{N}$. Now, by Khintchine's inequality, for $0 < p < \infty$ and $x_1, x_2, ..., x_m \in \mathbb{C}$ we have

$$A_p \left(\sum_{i=1}^{m} |x_i|^2\right)^{1/2} \leq \left(\mathbb{E}_{\boldsymbol{\sigma}}\left[\left|\sum_{i=1}^{m} x_i \sigma_i\right|^p\right]\right)^{1/p} \leq B_p \left(\sum_{i=1}^{m} |x_i|^2\right)^{1/2},$$

where $A_p, B_p > 0$ are constants depending on $p$. Sharp values for $A_p, B_p$ were found by Haagerup [1981]. In particular, for $p = 1$ he found that $A_1 = \frac{1}{\sqrt{2}}$ and $B_1 = 1$. By using Khintchine's inequality for each $\mathbb{E}_{\boldsymbol{\sigma}}\left[\left|\sum_{i=1}^{m\widehat{\boldsymbol{\mu}}_m(x)} \sigma_i\right|\right]$ with these constants, we get

$$\frac{1}{\sqrt{2}}\sqrt{m\widehat{\boldsymbol{\mu}}_m(x)} \leq \mathbb{E}_{\boldsymbol{\sigma}}\left[\left|\sum_{i=1}^{m\widehat{\boldsymbol{\mu}}_m(x)} \sigma_i\right|\right] \leq \sqrt{m\widehat{\boldsymbol{\mu}}_m(x)},$$

and hence

$$\frac{1}{2\sqrt{2}}\sum_{x\in\hat{S}}\sqrt{m\widehat{\boldsymbol{\mu}}_m(x)} \leq m\hat{\mathfrak{R}}_m(\boldsymbol{X}) \leq \frac{1}{2}\sum_{x\in\hat{S}}\sqrt{m\widehat{\boldsymbol{\mu}}_m(x)}.$$

Dividing by $m$ completes the proof. $\qquad\square$

Remark: We also give an exact expression for $\hat{\mathfrak{R}}_m(\boldsymbol{X})$ in Lemma A.1, and show in Corollary A.1 with a more delicate analysis that

$$\frac{\|\widehat{\boldsymbol{\mu}}_m\|_{1/2}^{1/2}}{\sqrt{2\pi m}} - \frac{3}{2}\sqrt{\frac{1}{2\pi}}\frac{1}{m^{3/2}}\left\|\widehat{\boldsymbol{\mu}}_m^+\right\|_{-1/2}^{-1/2} \leq \hat{\mathfrak{R}}_m(\boldsymbol{X}) \leq \frac{\|\widehat{\boldsymbol{\mu}}_m\|_{1/2}^{1/2}}{\sqrt{2\pi m}} + \sqrt{\frac{1}{2\pi}}\frac{1}{m^{3/2}}\left\|\widehat{\boldsymbol{\mu}}_m^+\right\|_{-1/2}^{-1/2}.$$

### 3.2 Proof of Theorem 2.2

The proof follows from applying the lower bound of Lemma 3.2 to the following lemma:

**Lemma 3.3** (lower bound by empirical Rademacher). *For all $m \in \mathbb{N}$, $\delta \in (0,1)$, and $\boldsymbol{\mu} \in \Delta_{\mathbb{N}}$, we have that*

$$\|\widehat{\boldsymbol{\mu}}_m - \boldsymbol{\mu}\|_{\mathsf{TV}} \geq \frac{1}{2}\hat{\mathfrak{R}}_m(\boldsymbol{X}) - 3\sqrt{\frac{\log\frac{2}{\delta}}{m}}$$

*holds with probability at least $1 - \delta$.*

*Proof.* The proof is closely based on [Wainwright, 2019, Proposition 4.12], which states: Let $\boldsymbol{Y} = (Y_1, \ldots, Y_m) \sim \boldsymbol{\nu}^m$ for some distribution $\boldsymbol{\nu}$ on $\mathcal{Z}$, let $\mathcal{G} \subseteq [-b, b]^{\mathcal{Z}}$ be a function class, and let $\boldsymbol{\sigma} = (\sigma_1, \ldots, \sigma_m) \sim \mathrm{Uniform}(\{-1, 1\}^m)$. Then

$$\sup_{g\in\mathcal{G}}\left|\mathbb{E}_{Y\sim\boldsymbol{\nu}}[g(Y)] - \frac{1}{m}\sum_{i=1}^{m}g(Y_i)\right| \geq \frac{1}{2}\mathbb{E}_{\boldsymbol{\sigma},\boldsymbol{Y}}\left[\sup_{g\in\mathcal{G}}\left|\frac{1}{m}\sum_{i=1}^{m}\sigma_i g(Y_i)\right|\right] - \frac{\sup_{g\in\mathcal{G}}|\mathbb{E}_{Y\sim\boldsymbol{\nu}}[g(Y)]|}{2\sqrt{m}} - \delta \quad (10)$$

holds with probability at least $1 - e^{-\frac{n\delta^2}{2b^2}}$. Plugging in $\boldsymbol{X}$ for $\boldsymbol{Y}$, $\boldsymbol{\mu}$ for $\boldsymbol{\nu}$, $\mathbb{N}$ for $\mathcal{Z}$, 1 for $b$, and $\mathcal{F} := \{\mathbb{I}_A | A \subseteq \mathbb{N}\} = \{0,1\}^{\mathbb{N}}$ for $\mathcal{G}$ in (10) together with observing that

$$\|\widehat{\boldsymbol{\mu}}_m - \boldsymbol{\mu}\|_{\mathsf{TV}} := \sup_{A\subseteq\mathbb{N}}(\boldsymbol{\mu}(A) - \widehat{\boldsymbol{\mu}}_m(A)) = \sup_{f\in\mathcal{F}}\left|\mathbb{E}_{X\sim\boldsymbol{\mu}}[f(X)] - \frac{1}{m}\sum_{i=1}^{m}f(X_i)\right|,$$

$$\mathbb{E}_{\boldsymbol{\sigma},\boldsymbol{X}}\left[\sup_{f\in\mathcal{F}}\left|\frac{1}{m}\sum_{i=1}^{m}\sigma_i f(X_i)\right|\right] \geq \mathfrak{R}_m, \quad \text{and} \quad \sup_{f\in\mathcal{F}}\left|\mathbb{E}_{X\sim\boldsymbol{\mu}}[f(X)]\right| = 1,$$

followed by some algebraic manipulation we get

$$\|\widehat{\boldsymbol{\mu}}_m - \boldsymbol{\mu}\|_{\mathsf{TV}} \geq \frac{1}{2}\mathfrak{R}_m - \frac{1}{2\sqrt{m}} - \sqrt{\frac{2\log\frac{2}{\delta}}{m}} \quad (11)$$

with probability at least $1 - \delta/2$. Applying McDiarmid's inequality to the $1/m$-bounded-differences function $\hat{\mathfrak{R}}_m(\boldsymbol{X})$ (similar to [Mohri et al., 2012, Eq. (3.14)]) we get:

$$\frac{1}{2}\mathfrak{R}_m \geq \frac{1}{2}\hat{\mathfrak{R}}_m(\boldsymbol{X}) - \frac{1}{2}\sqrt{\frac{\log\frac{2}{\delta}}{2m}} \quad (12)$$

with probability at least $1 - \delta/2$. To conclude the proof, combine (11) and (12) with the union bound to get:

$$\|\widehat{\boldsymbol{\mu}}_m - \boldsymbol{\mu}\|_{\mathsf{TV}} \geq \frac{1}{2}\hat{\mathfrak{R}}_m(\boldsymbol{X}) - \frac{1}{2\sqrt{m}} - \frac{1}{2}\sqrt{\frac{\log\frac{2}{\delta}}{2m}} - \sqrt{\frac{2\log\frac{2}{\delta}}{m}}$$

with probability at least $1 - \delta$, and use the fact $-\frac{1}{2\sqrt{m}} - \frac{1}{2}\sqrt{\frac{\log\frac{2}{\delta}}{2m}} - \sqrt{\frac{2\log\frac{2}{\delta}}{m}} \geq -3\sqrt{\frac{\log\frac{2}{\delta}}{m}}$ for all $m \in \mathbb{N}, \delta \in (0,1)$. $\qquad\square$

**Remark 3.1.** *We note that by using a more careful analysis, the constants of Theorem 2.2 can be improved to yield, under the same assumptions, $\|\widehat{\boldsymbol{\mu}}_m - \boldsymbol{\mu}\|_{\mathsf{TV}} \geq \frac{1}{2}\hat{\mathfrak{R}}_m(\boldsymbol{X}) - \frac{1}{4\sqrt{m}} - \frac{3}{2}\sqrt{\frac{\log\frac{2}{\delta}}{2m}}$ with probability at least $1 - \delta$.*

## 3.3 Proof of Theorem 2.3

Invoking Fubini's theorem, we write

$$\frac{1}{\sqrt{m}}\mathbb{E}\left[\|\widehat{\boldsymbol{\mu}}_m\|_{1/2}^{1/2}\right] = \frac{1}{m}\sum_{i=1}^{\infty}\mathop{\mathbb{E}}_{X \sim \mathrm{Bin}(m,\boldsymbol{\mu}(i))}\left[\sqrt{X}\right].$$

Since $X \in \{0,1,2,\ldots\}$, we have $\sqrt{X} \leq X$ and hence $\mathbb{E}\left[\sqrt{X}\right] \leq \mathbb{E}[X]$. On the other hand, Jensen's inequality implies $\mathbb{E}\left[\sqrt{X}\right] \leq \sqrt{\mathbb{E}[X]}$, whence

$$\frac{1}{\sqrt{m}}\mathbb{E}\left[\|\widehat{\boldsymbol{\mu}}_m\|_{1/2}^{1/2}\right] \leq \frac{1}{m}\sum_{i=1}^{\infty}\min\{\sqrt{m\boldsymbol{\mu}(i)}, m\boldsymbol{\mu}(i)\} \tag{13}$$

$$= \sum_{i:\,\boldsymbol{\mu}(i)\leq 1/m}\boldsymbol{\mu}(i) + \frac{1}{\sqrt{m}}\sum_{i:\,\boldsymbol{\mu}(i)>1/m}\sqrt{\boldsymbol{\mu}(i)} \leq 2\Lambda_m(\boldsymbol{\mu}). \tag{14}$$

The high-probability bound follows from applying McDiarmid's inequality to the $2/m$-bounded-differences function: for all $\delta \in (0,1)$, we have

$$\Phi_m(\widehat{\boldsymbol{\mu}}_m) \leq \mathbb{E}\left[\Phi_m(\widehat{\boldsymbol{\mu}}_m)\right] + \sqrt{\log(1/\delta)/m}.$$

$\qquad\square$

## 3.4 Statement and proof of the refined version of Theorem 2.5

**Theorem 3.1.** *There is a universal constant $C > 0$ such that for all $\Lambda \geq 2$ and $0 < \varepsilon, \delta < 1$, the MLE verifies the following optimality property: For all $\boldsymbol{\mu} \in \Delta_{\mathbb{N}}$ with $\|\boldsymbol{\mu}[2\varepsilon\delta/9]\|_{1/2} \leq \Lambda$, if $(X_1,\ldots,X_m) \sim \boldsymbol{\mu}^m$ and $m \geq \frac{C}{\varepsilon^2}\max\left\{\Lambda, \ln\delta^{-1}\right\}$, then $\|\widehat{\boldsymbol{\mu}}_m - \boldsymbol{\mu}\|_{\mathsf{TV}} < \varepsilon$ holds with probability at least $1 - \delta$.*

*On the other hand, for all $\Lambda \geq 2$ and $0 < \varepsilon < 1/16, 0 < \delta < 1$, for any estimator $\bar{\boldsymbol{\mu}}: \mathbb{N}^m \to \Delta_{\mathbb{N}}$ there is a $\boldsymbol{\mu} \in \Delta_{\mathbb{N}}$ with $\|\boldsymbol{\mu}[\varepsilon/18]\|_{1/2} \leq \Lambda$ such that $\bar{\boldsymbol{\mu}}$ must require at least $m \geq \frac{C}{\varepsilon^2}\Lambda$ samples in order for $\|\bar{\boldsymbol{\mu}} - \boldsymbol{\mu}\|_{\mathsf{TV}} < \varepsilon$ to hold with probability at least $3/4$, and for any estimator $\bar{\boldsymbol{\nu}}: \mathbb{N}^m \to \Delta_{\mathbb{N}}$ there is a $\boldsymbol{\nu} \in \Delta_{\mathbb{N}}$ with $\|\boldsymbol{\nu}[2\varepsilon\delta/9]\|_{1/2} \leq \Lambda$, such that $\bar{\boldsymbol{\nu}}$ must require at least $m \geq \frac{C}{\varepsilon^2}\ln\frac{1}{\delta}$ samples in order for $\|\bar{\boldsymbol{\nu}} - \boldsymbol{\nu}\|_{\mathsf{TV}} < \varepsilon$ to hold with probability at least $1 - \delta$.*

**Minimax risk.** For any $\Lambda \in [2,\infty), 0 < \varepsilon, \delta < 1$, we define the minimax risk

$$\mathcal{R}_m(\Lambda, \varepsilon, \delta) := \inf_{\bar{\boldsymbol{\mu}}} \sup_{\boldsymbol{\mu}:\|\boldsymbol{\mu}[2\varepsilon\delta/9]\|_{1/2}<\Lambda} \mathop{\mathbb{P}}_{\boldsymbol{X}\sim\boldsymbol{\mu}^m}\left(\|\bar{\boldsymbol{\mu}} - \boldsymbol{\mu}\|_{\mathsf{TV}} > \varepsilon\right),$$

where the infimum is taken over all functions $\bar{\boldsymbol{\mu}}: \mathbb{N}^m \to \Delta_{\mathbb{N}}$, and the supremum is taken over the subset of distributions such that $\|\boldsymbol{\mu}[2\varepsilon\delta/9]\|_{1/2} < \Lambda$.

**Upper bound.** Let $\Lambda \in [2, \infty), 0 < \varepsilon, \delta < 1, \boldsymbol{\mu} \in \Delta_{\mathbb{N}}$, such that $\|\boldsymbol{\mu}[2\varepsilon\delta/9]\|_{1/2} \leq \Lambda, m \in \mathbb{N}$, $(X_1, \ldots, X_m) \sim \boldsymbol{\mu}$ and let $\widehat{\boldsymbol{\mu}}_m$ be the MLE. For $\eta > 0$, consider the two truncated distributions $\boldsymbol{\mu}[\eta]$ and $\widehat{\boldsymbol{\mu}}'_m$, where we define the latter as

$$\widehat{\boldsymbol{\mu}}'_m(i) \quad := \quad \widehat{\boldsymbol{\mu}}_m(i)\mathbf{1}[\boldsymbol{\mu}[\eta](i) > 0], \qquad i \in \mathbb{N}.$$

By the triangle inequality, $\mathbb{P}\left(\|\widehat{\boldsymbol{\mu}}_m - \boldsymbol{\mu}\|_{\mathsf{TV}} > \varepsilon\right) \leq \mathbb{P}\left(\mathcal{E}_1 + \mathcal{E}_2 + \mathcal{E}_3 > \varepsilon\right)$, where

$$\mathcal{E}_1 := \left\|\widehat{\boldsymbol{\mu}}_m - \widehat{\boldsymbol{\mu}}'_m\right\|_{\mathsf{TV}}, \ \mathcal{E}_2 := \left\|\widehat{\boldsymbol{\mu}}'_m - \boldsymbol{\mu}[2\varepsilon\delta/9]\right\|_{\mathsf{TV}}, \ \mathcal{E}_3 := \left\|\boldsymbol{\mu}[2\varepsilon\delta/9] - \boldsymbol{\mu}\right\|_{\mathsf{TV}}.$$

By Markov's inequality,

$$\mathbb{P}\left(\mathcal{E}_1 > \frac{\varepsilon}{3}\right) \leq \frac{3}{\varepsilon}\mathbb{E}\left[\|\widehat{\boldsymbol{\mu}}_m - \widehat{\boldsymbol{\mu}}'_m\|_{\mathsf{TV}}\right] = \frac{3}{2\varepsilon}\mathbb{E}\left[\sum_{i=1}^{\infty}|\widehat{\boldsymbol{\mu}}_m(i) - \widehat{\boldsymbol{\mu}}'_m(i)|\right]$$

$$= \frac{3}{2\varepsilon}\mathbb{E}\left[\frac{1}{m}\sum_{i\in\mathbb{N}\,:\,\Pi_{\boldsymbol{\mu}}(i)>T_{\boldsymbol{\mu}}(\eta)}\sum_{t=1}^{m}\mathbf{1}[X_t = i]\right] = \frac{3}{2\varepsilon}\mathbb{P}\left(\Pi_{\boldsymbol{\mu}}(X_t) > T_{\boldsymbol{\mu}}(\eta)\right) \leq \frac{\delta}{3}.$$

Moreover, $\mathcal{E}_3 = \frac{1}{2}\sum_{i>T_{\boldsymbol{\mu}}(\eta)}^{\infty}\boldsymbol{\mu}^{\downarrow}(i) \leq \frac{\varepsilon\delta}{9} \leq \frac{\varepsilon}{3}$. In order to apply the union bound, it remains to handle $\mathbb{P}(\mathcal{E}_2 > \varepsilon/3)$. This is achieved in two standard steps. The first follows an argument similar to that of [Berend and Kontorovich, 2013, Lemma 5], that bounds from above the quantity in expectation using Jensen's inequality, $\mathbb{E}[\mathcal{E}_2] \leq \frac{\|\boldsymbol{\mu}[2\varepsilon\delta/9]\|_{1/2}^{1/2}}{\sqrt{m}} \leq \sqrt{\frac{\Lambda}{m}}$. An application of McDiarmid's inequality controls the fluctuations around the expectation [Berend and Kontorovich, 2013, (7.5)] and concludes the proof.

$\square$

**Sample complexity lower bound $m = \Omega\left(\frac{\log \delta^{-1}}{\varepsilon^2}\right)$.** See Lemma B.1.

**Sample complexity lower bound $m = \Omega\left(\frac{\Lambda}{\varepsilon^2}\right)$.** Let $\varepsilon \in (0, 1/16)$ and $\Lambda > 2$. First observe that $\Lambda/2 \leq 2\lfloor\Lambda/2\rfloor \leq \Lambda$, and $2\lfloor\Lambda/2\rfloor \in 2\mathbb{N}$. As a result,

$$\mathcal{R}_m(\Lambda, \varepsilon, \delta) \overset{(i)}{\geq} \inf_{\bar{\boldsymbol{\mu}}}\sup_{\boldsymbol{\mu}:\|\boldsymbol{\mu}[2\varepsilon\delta/9]\|_{1/2}\leq 2\lfloor\Lambda/2\rfloor}\mathbb{P}_{\boldsymbol{X}\sim\boldsymbol{\mu}^m}\left(\|\bar{\boldsymbol{\mu}} - \boldsymbol{\mu}\|_{\mathsf{TV}} > \varepsilon\right)$$

$$\overset{(ii)}{\geq} \inf_{\bar{\boldsymbol{\mu}}}\sup_{\boldsymbol{\mu}\in\Delta_{2\lfloor\Lambda/2\rfloor}}\mathbb{P}_{\boldsymbol{X}\sim\boldsymbol{\mu}^m}\left(\|\bar{\boldsymbol{\mu}} - \boldsymbol{\mu}\|_{\mathsf{TV}} > \varepsilon\right) \overset{(iii)}{\geq} \frac{1}{2}\left(1 - \frac{mC\varepsilon^2}{2\lfloor\Lambda/2\rfloor}\right) \geq \frac{1}{2}\left(1 - \frac{2mC\varepsilon^2}{\Lambda}\right)$$

where $(i)$ and $(ii)$ follow from taking the supremum over increasingly smaller sets, $(iii)$ is Lemma B.2 invoked for $2\lfloor\Lambda/2\rfloor \in \mathbb{N}$, and $C > 0$ is a universal constant. To conclude, $m \leq \frac{\Lambda}{4C\varepsilon^2} \implies \mathcal{R}_m(\Lambda, \varepsilon, \delta) \geq 1/4$, which yields the second lower bound. $\square$

Remark: The universal constant in the lower bound obtained by Tsybakov's method at Lemma B.2 is suboptimal, and we give a short proof in the appendix for completeness. We refer the reader to the more involved methods of Kamath et al. [2015] for obtaining tighter bounds.

### 3.5 Proof of Theorem 2.4

Let $d \in 2\mathbb{N}$ and $m \in \mathbb{N}$, and restrict the problem to $\boldsymbol{\mu} \in \Delta_d$. Let $\varepsilon \in (0, 1/16)$. By Lemma B.2, $\bar{\mathcal{R}}_m(d, \varepsilon) := \inf_{\bar{\boldsymbol{\mu}}}\sup_{\boldsymbol{\mu}\in\Delta_d}\mathbb{P}\left(\|\bar{\boldsymbol{\mu}} - \boldsymbol{\mu}\|_{\mathsf{TV}} > \varepsilon\right) \geq \frac{1}{2}\left(1 - \frac{Cm\varepsilon^2}{d}\right)$ for some $C > 0$, whence Markov's inequality yields

$$\frac{1}{2}\left(1 - \frac{Cm\varepsilon^2}{d}\right) \leq \frac{1}{\varepsilon}\inf_{\bar{\boldsymbol{\mu}}}\sup_{\boldsymbol{\mu}\in\Delta_d}\mathbb{E}\left[\|\bar{\boldsymbol{\mu}} - \boldsymbol{\mu}\|_{\mathsf{TV}}\right].$$

Restrict $m \geq \frac{d}{b^2}$, with $b := \sqrt{3C/16}$ and set $\varepsilon = \sqrt{\frac{d}{3Cm}}$, so that

$$\inf_{\bar{\boldsymbol{\mu}}}\sup_{\boldsymbol{\mu}\in\Delta_d}\mathbb{E}\left[\|\bar{\boldsymbol{\mu}} - \boldsymbol{\mu}\|_{\mathsf{TV}}\right] \geq \frac{1}{a}\sqrt{\frac{d}{m}}, \text{ where } a := \sqrt{27C} \tag{15}$$

Suppose that $\theta(\sqrt{d/m}) < \frac{1}{a}\sqrt{\frac{d}{m}}$, then by hypothesis,

$$\inf_{\bar{\boldsymbol{\mu}}} \sup_{\boldsymbol{\mu} \in \Delta_d} \mathbb{E}\left[\|\bar{\boldsymbol{\mu}} - \boldsymbol{\mu}\|_{\mathsf{TV}}\right] \leq \sup_{\boldsymbol{\mu} \in \Delta_d} \mathbb{E}\left[\Psi_m(\tilde{\boldsymbol{\mu}}_m)\right] \leq \sup_{\boldsymbol{\mu} \in \Delta_d} \theta\left(\mathbb{E}\left[\Phi_m\right]\right).$$

For $\boldsymbol{\mu} \in \Delta_d$, $\mathbb{E}\left[\sqrt{\frac{\|\hat{\boldsymbol{\mu}}_m\|_{1/2}}{m}}\right] \leq \sqrt{\frac{d}{m}}$. It follows that

$$\sup_{\boldsymbol{\mu} \in \Delta_d} \theta\left(\mathbb{E}\left[\Phi_m\right]\right) \leq \theta\left(\sqrt{\frac{d}{m}}\right) < \frac{1}{a}\sqrt{\frac{d}{m}},$$

which contradicts (15). We have therefore established, for

$$r \in R := \left\{\sqrt{d/m} \colon (m, d) \in \mathbb{N} \times 2\mathbb{N}, m \geq \frac{d}{b^2}\right\},$$

the lower bound $\theta(r) \geq r/a$. We extend the lower bound to the open interval $(0, b)$, by observing that $R$ is dense in $(0, b)$ followed by a continuity argument. $\qquad\square$

## Broader Impact

This work is of purely theoretical nature and does not present any foreseeable societal consequence.

## Acknowledgments and Disclosure of Funding

We are thankful to Clayton Scott for the insightful conversations, and to the anonymous referees for their valuable comments. This research was partially supported by the Israel Science Foundation (grant No. 1602/19) and Google Research.

## Footnotes

[1] While $\boldsymbol{\mu}^{\downarrow}$ is uniquely defined, $\Pi_{\boldsymbol{\mu}}^{\downarrow}$ is not. Uniqueness could be ensured by taking the lexicographically first permutation, but will not be needed for our results.

[2] The half-norm is not a proper vector-space norm, as it lacks sub-additivity.

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
