[Supplementary Material]

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

# A  Analysis of the Empirical Rademacher complexity

From Lemma 3.2 (see also [Scott and Nowak, 2006, Section 7.1, Appendix E.]), we see that the Khintchine inequality already yields a control of $\hat{\mathfrak{R}}_m(\boldsymbol{X})$ by $\|\widehat{\boldsymbol{\mu}}_m\|_{1/2}^{1/2}$ up to universal constants.

$$\frac{1}{2\sqrt{2}}\|\widehat{\boldsymbol{\mu}}_m\|_{1/2}^{1/2} \leq \hat{\mathfrak{R}}_m(\boldsymbol{X}) \leq \frac{1}{2}\|\widehat{\boldsymbol{\mu}}_m\|_{1/2}^{1/2}.$$

Furthermore, it is possible to derive an exact expression for it, from the expected absolute distance of a symmetric random walk:

**Lemma A.1** (Empirical Rademacher complexity, exact expression). *Let $\boldsymbol{X} = (X_1, \ldots, X_m)$ and let $\widehat{\boldsymbol{\mu}}_m$ be the empirical measure constructed from the sample $\boldsymbol{X}$. Then,*

$$\hat{\mathfrak{R}}_m(\boldsymbol{X}) = \frac{1}{m}\sum_{x:\,\widehat{\boldsymbol{\mu}}_m(x)>0}\frac{1}{2^{m\widehat{\boldsymbol{\mu}}_m(x)}}\left\lceil\frac{m\widehat{\boldsymbol{\mu}}_m(x)}{2}\right\rceil\binom{m\widehat{\boldsymbol{\mu}}_m(x)}{\lceil m\widehat{\boldsymbol{\mu}}_m(x)/2\rceil}.$$

*Proof.* Write $m\hat{\mathfrak{R}}_m(\boldsymbol{X}) = \sum_{x:\,\widehat{\boldsymbol{\mu}}_m(x)>0}\frac{1}{2}\mathbb{E}_{\boldsymbol{\sigma}}\left[\left|\sum_{i=1}^{m\widehat{\boldsymbol{\mu}}_m(x)}\sigma_i\right|\right]$ as in the proof of Lemma 3.2.
Now, observe that $\mathbb{E}_{\boldsymbol{\sigma}}\left[\left|\sum_{i=1}^{m\widehat{\boldsymbol{\mu}}_m(x)}\sigma_i\right|\right]$ is the expectation value of the absolute distance of a 1-dimensional symmetric random walk after $m\widehat{\boldsymbol{\mu}}_m(x)$ steps, also known as the "heads minus tails" process [Handelsman, 1991]:

$$\mathbb{E}_{\boldsymbol{\sigma}}\left[\left|\sum_{i=1}^{m\widehat{\boldsymbol{\mu}}_m(x)}\sigma_i\right|\right] = \frac{1}{m2^{m\widehat{\boldsymbol{\mu}}_m(x)}}\left\lceil\frac{m\widehat{\boldsymbol{\mu}}_m(x)}{2}\right\rceil\binom{m\widehat{\boldsymbol{\mu}}_m(x)}{\lceil m\widehat{\boldsymbol{\mu}}_m(x)/2\rceil}.$$

$\square$

However, the above is inconvenient and involves the computation of factorials. Leveraging delicate bounds for the central binomial coefficient obtained with the Wallis product in Dunbar [2009], we derive the following corollary, that gives exact the first-order constant in terms of the *half-norm*, makes the *minus-half-norm* appear as a second dominant term, and that is easily computable.

**Corollary A.1** (Empirical Rademacher complexity, first order bound). *Let $\boldsymbol{X} = (X_1, \ldots, X_m)$ and let $\widehat{\boldsymbol{\mu}}_m$ be the empirical measure constructed from the sample $\boldsymbol{X}$. Then writing*

$$\phi_m(\boldsymbol{X}) := \frac{\|\widehat{\boldsymbol{\mu}}_m\|_{1/2}^{1/2}}{\sqrt{2\pi m}},$$

*it holds that*

$$\frac{\|\widehat{\boldsymbol{\mu}}_m\|_{1/2}^{1/2}}{\sqrt{2\pi m}} - \frac{3}{2}\sqrt{\frac{1}{2\pi}}\frac{1}{m^{3/2}}\left\|\widehat{\boldsymbol{\mu}}_m^+\right\|_{-1/2}^{-1/2} \leq \hat{\mathfrak{R}}_m(\boldsymbol{X}) \leq \frac{\|\widehat{\boldsymbol{\mu}}_m\|_{1/2}^{1/2}}{\sqrt{2\pi m}} + \sqrt{\frac{1}{2\pi}}\frac{1}{m^{3/2}}\left\|\widehat{\boldsymbol{\mu}}_m^+\right\|_{-1/2}^{-1/2}.$$

*Proof.* Let $n \in \mathbb{N}$, if $n = 2k, k \geq 1$,

$$\frac{1}{2^n}\left\lceil\frac{n}{2}\right\rceil\binom{n}{\lceil n/2\rceil} = \frac{1}{4^k}k\binom{2k}{k},$$

and if $n = 2k - 1, k \geq 1$, $\lceil n/2\rceil = k$ such that similarly,

$$\frac{1}{2^n}\left\lceil\frac{n}{2}\right\rceil\binom{n}{\lceil n/2\rceil} = \frac{1}{2^{2k-1}}k\binom{2k-1}{k} = \frac{2}{4^k}k\frac{(2k-1)!}{k!(2k-k-1)!}$$

$$= \frac{2}{4^k}k\frac{(2k)!(2k-k)}{(2k)k!(2k-k)!} = \frac{2}{4^k}k\frac{2k-k}{2k}\binom{2k}{k} = \frac{1}{4^k}k\binom{2k}{k}.$$

Moreover, from Dunbar [2009, p.11], for $k \geq 1$, an application of the Wallis product yields,

$$\frac{k}{\sqrt{\pi/2}\sqrt{2k+1}}\left(1 - \frac{1}{2k}\right) \leq \frac{1}{4^k}k\binom{2k}{k} \leq \frac{k}{\sqrt{\pi/2}\sqrt{2k+1}}\left(1 + \frac{1}{2k}\right).$$

If follows that when $n = 2k$,

$$\sqrt{\frac{n}{2\pi}}\left\{\sqrt{\frac{n}{n+1}}\left(1-\frac{1}{n}\right)\right\} \le \frac{1}{2^n}\left\lceil\frac{n}{2}\right\rceil\binom{n}{\lceil n/2\rceil} \le \sqrt{\frac{n}{2\pi}}\left\{\sqrt{\frac{n}{n+1}}\left(1+\frac{1}{n}\right)\right\},$$

and for $n = 2k - 1$,

$$\sqrt{\frac{n}{2\pi}}\left\{\frac{n+1}{\sqrt{n(n+2)}}\left(1-\frac{1}{n+1}\right)\right\} \le \frac{1}{2^n}\left\lceil\frac{n}{2}\right\rceil\binom{n}{\lceil n/2\rceil} \le \sqrt{\frac{n}{2\pi}}\left\{\frac{n+1}{\sqrt{n(n+2)}}\left(1+\frac{1}{n+1}\right)\right\}$$

For all $n \in \mathbb{N}$,

$$\sqrt{\frac{n}{n+1}}\left(1+\frac{1}{n}\right) \le \frac{n+1}{\sqrt{n(n+2)}}\left(1+\frac{1}{n+1}\right) \le 1+\frac{1}{n},$$

$$\sqrt{\frac{n}{n+1}}\left(1-\frac{1}{n}\right) \ge \frac{n+1}{\sqrt{n(n+2)}}\left(1-\frac{1}{n+1}\right) \ge 1-\frac{3}{2n},$$

such that

$$\hat{\mathfrak{R}}_m(\boldsymbol{X}) \le \sqrt{\frac{1}{2\pi m}}\sum_{x\,:\,\widehat{\boldsymbol{\mu}}_m(x)>0}\sqrt{\widehat{\boldsymbol{\mu}}_m(x)}\left\{1+\frac{1}{m\widehat{\boldsymbol{\mu}}_m(x)}\right\}$$

$$\le \phi_m(\boldsymbol{X}) + \sqrt{\frac{1}{2\pi}}\frac{1}{m^{3/2}}\left\|\widehat{\boldsymbol{\mu}}_m^+\right\|_{-1/2}^{-1/2},$$

where we wrote

$$\left\|\widehat{\boldsymbol{\mu}}_m^+\right\|_{-1/2}^{-1/2} := \sum_{x\in\mathbb{N}}\frac{\mathbf{1}[\widehat{\boldsymbol{\mu}}_m(x)>0]}{\sqrt{\widehat{\boldsymbol{\mu}}_m(x)}},$$

and conversely,

$$\hat{\mathfrak{R}}_m(\boldsymbol{X}) \ge \phi_m(\boldsymbol{X}) - \frac{3}{2}\sqrt{\frac{1}{2\pi}}\frac{1}{m^{3/2}}\left\|\widehat{\boldsymbol{\mu}}_m^+\right\|_{-1/2}^{-1/2}.$$

$\square$

## B  Auxiliary lemmas for lower bounds

**Lemma B.1** (Sample complexity lower bound $m = \Omega\left(\log\delta^{-1}/\varepsilon^2\right)$)**.** *Let* $\Lambda \ge 2$, $0 < \varepsilon < 1/2$ *and* $0 < \delta < 1$. *For* any *estimator* $\bar{\boldsymbol{\nu}}\colon \mathbb{N}^m \to \Delta_{\mathbb{N}}$ *there is a* $\boldsymbol{\nu} \in \Delta_{\mathbb{N}}$ *with* $\|\boldsymbol{\nu}[2\varepsilon\delta/9]\|_{1/2} \le \Lambda$, *such that* $\bar{\boldsymbol{\nu}}$ *must require at least* $m = \Omega\left(\frac{\log\delta^{-1}}{\varepsilon^2}\right)$ *samples in order for* $\|\bar{\boldsymbol{\nu}} - \boldsymbol{\nu}\|_{\mathrm{TV}} < \varepsilon$ *to hold with probability at least* $1 - \delta$.

*Proof.* The proof is standard and consists of lower bounding the difficulty of learning a biased coin. Recall that for $\boldsymbol{\mu}_0 := (1/2, 1/2), \boldsymbol{\mu}_\varepsilon := (1/2 - \varepsilon, 1/2 + \varepsilon)$, direct computations lead to $\|\boldsymbol{\mu}_0 - \boldsymbol{\mu}_\varepsilon\|_1 = 2\varepsilon$, and $D_{\mathrm{KL}}\left(\boldsymbol{\mu}_\varepsilon\|\boldsymbol{\mu}_0\right) = (1/2 - \varepsilon)\ln\frac{1/2-\varepsilon}{1/2} + (1/2 + \varepsilon)\ln\frac{1/2+\varepsilon}{1/2} \le 4\varepsilon^2$, where $D_{\mathrm{KL}}\left(\boldsymbol{\mu}_\varepsilon\|\boldsymbol{\mu}_0\right)$ is the KL divergence between $\boldsymbol{\mu}_\varepsilon$ and $\boldsymbol{\mu}_0$. We also verify that $\|\boldsymbol{\mu}_\varepsilon\|_{1/2} \le \|\boldsymbol{\mu}_0\|_{1/2} \le 2 \le \Lambda$, hence also for their truncated version. From an immediate corollary of LeCam's theorem [Tsybakov, 2009, Theorem 2.2, Lemma 2.6], $\mathcal{R}_m(\Lambda, \varepsilon, \delta) \ge \frac{1}{2}\exp\left(-mD_{\mathrm{KL}}\left(\boldsymbol{\mu}_\varepsilon\|\boldsymbol{\mu}_0\right)\right)$, whence $m \le \frac{1}{4\varepsilon^2}\log\frac{\delta^{-1}}{2} \implies \mathcal{R}_m(\Lambda, \varepsilon, \delta) \ge \delta$. $\square$

**Lemma B.2.** *Let* $d \in 2\mathbb{N}, d \ge 16, m \in \mathbb{N}, \varepsilon \in (0, 1/16)$, *and let*

$$\bar{\mathcal{R}}_m(d, \varepsilon) := \inf_{\bar{\boldsymbol{\mu}}}\sup_{\boldsymbol{\mu}:\boldsymbol{\mu}\in\Delta_d}\mathop{\mathbb{P}}_{\boldsymbol{X}\sim\boldsymbol{\mu}^m}\left(\|\bar{\boldsymbol{\mu}} - \boldsymbol{\mu}\|_{\mathrm{TV}} > \varepsilon\right),$$

*where the infimum is taken over all* $\bar{\boldsymbol{\mu}}\colon [d]^m \to \Delta_d$. *Then there is a universal* $C > 0$ *such that*

$$\bar{\mathcal{R}}_m(d, \varepsilon) \ge \frac{1}{2}\left(1 - \frac{Cm\varepsilon^2}{d}\right).$$

*Proof.* As is customary in Analysis, the universal constant $C > 0$ may change its value from expression to expression. Consider the family of distributions

$$\mathcal{D}(d) := \left\{ \boldsymbol{\mu}^{(\boldsymbol{\sigma})} \in \Delta_d, \boldsymbol{\sigma} \in \{0, 1\}^{d/2} \right\},$$

where

$$\boldsymbol{\mu}^{(\boldsymbol{\sigma})} := \frac{1}{d} \left( 1 + 16\varepsilon\sigma_1, 1 - 16\varepsilon\sigma_1, 1 + 16\varepsilon\sigma_2, 1 - 16\varepsilon\sigma_2, \ldots, 1 + 16\varepsilon\sigma_{d/2}, 1 - 16\varepsilon\sigma_{d/2} \right).$$

From the Varshamov-Gilbert bound [Tsybakov, 2009, Lemma 2.9], there exists a $\tilde{\mathcal{D}}(d) \subsetneq \mathcal{D}(d)$ satisfying $(a)$ $\left| \tilde{\mathcal{D}}(d) \right| > 2^{d/16}$, $(b)$ for $\boldsymbol{\mu}^{(\boldsymbol{\sigma})}, \boldsymbol{\mu}^{(\boldsymbol{\sigma}')} \in \tilde{\mathcal{D}}(d), \boldsymbol{\sigma} \neq \boldsymbol{\sigma}' \implies \left\| \boldsymbol{\mu}^{(\boldsymbol{\sigma})} - \boldsymbol{\mu}^{(\boldsymbol{\sigma}')} \right\|_{\text{TV}} \geq 2\varepsilon$, and $(c)$ $\boldsymbol{\mu}^{(\boldsymbol{0})} \in \tilde{\mathcal{D}}(d)$. It is straightforward to verify that $D_{\text{KL}} \left( \boldsymbol{\mu}^{(\boldsymbol{\sigma})} \| \boldsymbol{\mu}^{(\boldsymbol{0})} \right) \leq C\varepsilon^2$. Applying Tsybakov's method [Tsybakov, 2009, Theorem 2.5],

$$\bar{\mathcal{R}}_m(d, \varepsilon) \geq \inf_{\bar{\boldsymbol{\mu}}} \sup_{\boldsymbol{\mu} \in \tilde{\mathcal{D}}(d)} \mathbb{P}\left( \| \bar{\boldsymbol{\mu}} - \boldsymbol{\mu} \|_{\text{TV}} > \varepsilon \right)$$

$$\geq \frac{1}{2} \left( 1 - \frac{\frac{4m}{|\tilde{\mathcal{D}}(d)|} \sum_{\boldsymbol{\mu}^{(\boldsymbol{\sigma})} \in \tilde{\mathcal{D}}(d)} D_{\text{KL}} \left( \boldsymbol{\mu}^{(\boldsymbol{\sigma})} \| \boldsymbol{\mu}^{(\boldsymbol{0})} \right)}{\ln \left| \tilde{\mathcal{D}}(d) \right|} \right)$$

so that $\bar{\mathcal{R}}_m(d, \varepsilon) \geq \frac{1}{2} \left( 1 - \frac{Cm\varepsilon^2}{d} \right)$. $\qquad\square$

## C  Convergence properties of the empirical bound

In this section, we briefly analyze convergence of $\frac{1}{\sqrt{m}} \| \widehat{\boldsymbol{\mu}}_m \|_{1/2}^{1/2}$. In Proposition C.1 we confirm that the quantity converges *almost surely* and in $L_1$, but with Proposition C.2, with show that this convergence can be *arbitrarily slow*.

**Proposition C.1** ($L_1$ and almost sure convergence). *Let $\boldsymbol{\mu} \in \Delta_{\mathbb{N}}$ and let $\boldsymbol{X} := (X_1, \ldots, X_m) \sim \boldsymbol{\mu}^m$. Then, $\frac{1}{\sqrt{m}} \| \widehat{\boldsymbol{\mu}}_m \|_{1/2}^{1/2} \xrightarrow{L_1} 0$ and $\frac{1}{\sqrt{m}} \| \widehat{\boldsymbol{\mu}}_m \|_{1/2}^{1/2} \xrightarrow{a.s.} 0$.*

*Proof.* For $L_1$ convergence the proof is as follows:

$$\lim_{m \to \infty} \mathbb{E}\left[ \left| \frac{1}{\sqrt{m}} \| \widehat{\boldsymbol{\mu}}_m \|_{1/2}^{1/2} - 0 \right| \right] = \lim_{m \to \infty} \mathbb{E}\left[ \frac{1}{\sqrt{m}} \| \widehat{\boldsymbol{\mu}}_m \|_{1/2}^{1/2} \right]$$

$$\leq \lim_{m \to \infty} 2\Lambda_m(\boldsymbol{\mu}) \qquad\qquad \text{(Theorem 2.3)}$$

$$= 0. \qquad\qquad \text{([Berend and Kontorovich, 2013, Lemma 7])}$$

Now, for almost sure convergence, recall that $\frac{1}{\sqrt{m}} \| \widehat{\boldsymbol{\mu}}_m \|_{1/2}^{1/2}$ satisfies $2/m$-bounded-differences. By the $L_1$ convergence established above, we have that for all $\varepsilon > 0$ there is an $M_\varepsilon \in \mathbb{N}$ s.t. for all $m \geq M_\varepsilon$, we have $\mathbb{E}\left[ \frac{1}{\sqrt{m}} \| \widehat{\boldsymbol{\mu}}_m \|_{1/2}^{1/2} \right] \leq \varepsilon/2$. Invoking McDiarmid's inequality, for every $m \geq M_\varepsilon$, we have

$$\mathbb{P}\left( \frac{1}{\sqrt{m}} \| \widehat{\boldsymbol{\mu}}_m \|_{1/2}^{1/2} \geq \varepsilon/2 \right) \leq \exp\left( -\frac{m\varepsilon^2}{2} \right).$$

Thus,

$$\sum_{m=1}^{\infty} \mathbb{P}\left( \frac{1}{\sqrt{m}} \| \widehat{\boldsymbol{\mu}}_m \|_{1/2}^{1/2} \geq \varepsilon/2 \right) \leq M_\varepsilon + \sum_{m=M_\varepsilon}^{\infty} \exp\left( -\frac{m\varepsilon^2}{2} \right) < \infty.$$

An application of the Borel-Cantelli lemma completes the proof:

$$\frac{1}{\sqrt{m}} \| \widehat{\boldsymbol{\mu}}_m \|_{1/2}^{1/2} \xrightarrow{a.s.} 0.$$

$\qquad\square$

To formalize our idea of arbitrarily slow convergence, we adapt the terminology developed in Deutsch and Hundal [2010a,b]. We begin with the set of all $[0,1]$-valued sequences that converge to 0:

$$\mathcal{U} := \{U \in [0,1]^{\mathbb{N}} : \lim_{m \to \infty} U(m) = 0\}.$$

Following Deutsch and Hundal [2010a, Definition 2.7], we will say that the statistic $\hat{\theta}_m : \mathbb{N}^m \to [0,1]$ *converges arbitrarily slowly to 0 in $L_1$* if

1. $\forall \mu \in \Delta_{\mathbb{N}}, \lim_{m \to \infty} \mathbb{E}\left[\hat{\theta}_m\right] = 0,$

2. $\forall U \in \mathcal{U}, \exists \mu \in \Delta_{\mathbb{N}}$ such that $\forall m \in \mathbb{N}, \mathbb{E}\left[\hat{\theta}_m\right] \geq U(m).$

It turns out [Deutsch and Hundal, 2010b, Remark 2.8, Theorem 2.9] that restricting the set $\mathcal{U}$ to the *decreasing* sequences,

$$\mathcal{U}^{\downarrow} := \{U \in [0,1]^{\mathbb{N}} : \sup_{m \in \mathbb{N}} U(m+1)/U(m) \leq 1, \lim_{m \to \infty} U(m) = 0\},$$

does not change the above definition of arbitrarily slow convergence.

**Proposition C.2** (Arbitrary slow convergence in $L_1$). *For any sequence $1 > r_1 > r_2 > \ldots$ decreasing to 0, there is a distribution $\boldsymbol{\mu} \in \Delta_{\mathbb{N}}$ such that $2\mathbb{E}_{\boldsymbol{X} \sim \mu^m}\left[\|\boldsymbol{\mu} - \widehat{\boldsymbol{\mu}}_m\|_{\mathrm{TV}}\right] > r_m$ for all $m \geq 1$.*

*Proof.*

$$2\mathbb{E}\left[\|\boldsymbol{\mu} - \widehat{\boldsymbol{\mu}}_m\|_{\mathrm{TV}}\right] = \mathbb{E}\left[\|\boldsymbol{\mu} - \widehat{\boldsymbol{\mu}}_m\|_1\right]$$

$$= \sum_{i=1}^{\infty} \mathbb{E}\left[|\boldsymbol{\mu}(i) - \widehat{\boldsymbol{\mu}}_m(i)|\right]$$

$$= \sum_{i=1}^{\infty} \mathbb{E}\left[\mathbf{1}[\widehat{\boldsymbol{\mu}}_m(i) > 0]\,|\boldsymbol{\mu}(i) - \widehat{\boldsymbol{\mu}}_m(i)| + \mathbf{1}[\widehat{\boldsymbol{\mu}}_m(i) = 0]\,|\boldsymbol{\mu}(i) - \widehat{\boldsymbol{\mu}}_m(i)|\right]$$

$$= \sum_{i=1}^{\infty} \mathbb{E}\left[\mathbf{1}[\widehat{\boldsymbol{\mu}}_m(i) > 0]\,|\boldsymbol{\mu}(i) - \widehat{\boldsymbol{\mu}}_m(i)|\right] + \sum_{i=1}^{\infty} \mathbb{E}\left[\mathbf{1}[\widehat{\boldsymbol{\mu}}_m(i) = 0]\boldsymbol{\mu}(i)\right]$$

$$= \sum_{i=1}^{\infty} \mathbb{E}\left[\mathbf{1}[\widehat{\boldsymbol{\mu}}_m(i) > 0]\,|\boldsymbol{\mu}(i) - \widehat{\boldsymbol{\mu}}_m(i)|\right] + \mathbb{E}\left[\sum_{i:\,\widehat{\boldsymbol{\mu}}_m(i)=0} \boldsymbol{\mu}(i)\right]$$

$$\geq \mathbb{E}\left[\sum_{i:\,\widehat{\boldsymbol{\mu}}_m(i)=0} \boldsymbol{\mu}(i)\right] = \mathbb{E}\left[\boldsymbol{\mu}\left(\mathbb{N} \setminus \{X_1, \ldots X_m\}\right)\right]$$

$$= \mathbb{E}\left[U_m\right],$$

where $U_m := \boldsymbol{\mu}\left(\mathbb{N} \setminus \{X_1, \ldots X_m\}\right)$ is the missing mass random variable. From [Berend and Kontorovich, 2012, Proposition 4], we have that: For any sequence $1 > r_1 > r_2 > \ldots$ decreasing to 0, there is a distribution $\boldsymbol{\mu} \in \Delta_{\mathbb{N}}$ such that $\mathbb{E}\left[U_m\right] > r_m$ for all $m \geq 1$. □

**Remark C.1.** *To our knowledge, the above result is the first to establish a connection between the TV risk $\|\boldsymbol{\mu} - \widehat{\boldsymbol{\mu}}_m\|_{\mathrm{TV}}$ and the missing mass $U_m$.*