[Reviews · NeurIPS 2020]

Review 1

Summary and Contributions: This paper considers the problem of learning a discrete distribution in total variation distance, when the support of the distribution may be infinite. It is a classical result that a priori, this problem is impossible without structural assumptions about the underlying distribution. This paper circumvents this difficulty by demonstrating a bound based on quantity which depends on the samples, namely, the 1/2 norm of the empirical distribution. This recovers the standard risk bounds with finite support, and moreover, they demonstrate that this bound is tight. They also (sort of) demonstrate a bound which approaches an instance-optimal bound, which also suggests that this quantity always is tight for every distribution. They prove their bound in two steps. First, they demonstrate that the empirical Rademacher complexity is another bound for the risk via tools from VC theory, and then they demonstrate that the 1/2 norm of the empirical distribution is always a constant factor approximation to this quantity, using some metric geometry. EDIT: I have read the rebuttal and my evaluation of the paper is unchanged.

Strengths: - I may be missing context, as this result is quite statistical in nature, and somewhat out of my field, but I find this result to be quite nice. It demonstrates tight rates for a very natural problem. - The proofs are very clean and elementary. - The equivalence between the empirical Rademacher complexity and the 1/2 norm is also quite nice and was somewhat surprising to me, even if it is not so difficult to prove.

Weaknesses: - The nature of their bound does not allow a user to obtain risk bounds a priori before seeing the samples, however, as they explain, this is necessary in their setting.

Correctness: I verified the proof of their main claim Theorem 2.1 carefully and I believe it is correct. I did not check the other proofs as carefully but they seem correct to me as well.

Clarity: I found that the paper is clearly presented and was enjoyable to read.

Relation to Prior Work: I am not an expert in this area but I believe it addressed it adequately.

Reproducibility: Yes

Additional Feedback: - I would be curious if any of this technology could be combined with shape-constrained technology, for instance, in settings where the shape-constraint is still itself insufficient to obtain convergent rates. - Why is the main result presented in terms of the 1/2-norm instead of the empirical Rademacher complexity, given that one result is that these two are equivalent?


Review 2

Summary and Contributions: The paper studies the problem of distribution learning on discrete domain distributions focusing on total variation as the metric for learning. To learn a distribution supported on d elements to epsilon-TV, it is well-known that Theta(d/epsilon^2) iid samples from it are both necessary and sufficient. The current paper explores the setting of when d is unknown and can be potentially infinite. They give a fully empirical quantity which captures tightly (up to constants) how accurate of an approximation the empirical distribution is to the true distribution (in terms of the TV between them). They show that this quantity is instance-optimal as well.

Strengths: The paper studies a classic question in learning theory with a long line of work and importance to many related questions. They present results which are a novel purely empirical characterization of learning from distributions when their support size is unknown. Such a characterization is novel and useful for practitioners who may not have much prior knowledge about the distribution they are trying to learn. The subject of the paper is relevant to the NeurIPS community and contains novel and significant results. Moreover the novel complexity measure proposed and analyzed in the paper has instance-optimal properties and some analysis of its convergence as m-> infinity is also provided.

Weaknesses: No significant weaknesses in my opinion. -- Post-author rebuttal and discussion-- Thank you to the authors and the other reviewers for the rebuttal and discussion around the relation of this work to prior works (Berend and Kontorovich, Kamath et al). In light of the discussion, I am revising my score down to a 7. I still feel the paper is strong enough for acceptance and advocate for this outcome.

Correctness: I have not verified all the proofs but they look correct. I have verified some of the main claims to be correct.

Clarity: Yes

Relation to Prior Work: Yes

Reproducibility: Yes

Additional Feedback: Comments: 1. I enjoyed reading the paper. It is well-written and the proofs are succinct.
 Questions: 1. Does the complexity measure proposed generalize/apply to settings where we want to learn distributions with some underlying structure? 2. Is there something that can be thought of as a crude analogue of the measure for continuous measures? Minor Typos and clarifications: 1. Line 69, function class definition can made a bit more clear. Perhaps write as N -> {0,1}^N?


Review 3

Summary and Contributions: =================== Post rebuttal ======================= I appreciate the authors for their feedback. My score stays unchanged for the following reasons. (i) It is possible to get a result that works for "infinite support" using Theorem 1 in Berend-Kontorovich 2013. The example in the authors' feedback does not address this. (ii) Whether the bound is "fully empirical" should not be the issue here. If one can bound |x-y| by a function of x, say, c*\sqrt{x}, there should be a similar bound in terms of y. Here, x can be the actual probability value, and y can be its empirical estimate. (iii) As for motivation, the paper is missing an important reference: "doubly-competitive distribution estimation" by Hao and Orlitsky, 2019, which is an extension of Orlitsky et al. (Good-Turing, 2015). The main contribution of the work appears to be "making the error bound fully empirical/adaptive." ===================================================== The paper studies the classical problem of estimating an unknown discrete distribution from an iid sample. The learning algorithm is simply the empirical (distribution) estimator, shown to possess an adaptive expected L1 error guarantee.

Strengths: Prior works have determined the exact behavior of the empirical estimator under the min-max formulation. The strength of this work is that it shows an adaptive estimation guarantee based on the counting statistics. For some distribution families, one should observe improvements over the known results.

Weaknesses: There are several weaknesses. The first one is that the empirical distribution estimator is often suboptimal, which is unsurprising. Hence, modern research works on distribution estimation have focused on how to design estimators that are more efficient than the empirical ones. Therefore, the problem itself is not that interesting, especially given the results of Orlitsky et al. (Good-Turing, 2015) and Valiant and Valiant (Instance-optimal, 2016). The second one is that the results are similar to those in some intermediary steps of the prior min-max work. Take the paper "On learning distributions from their samples" by Kamath et al. 2015 as an example. On page 20 of that paper (top part), the authors establish an O(\sqrt{(d-1)/m}) upper bound on the L1 estimation error of the empirical distribution. Just before this, there is an adaptive bound of roughly O(\sum_i \sqrt{\mu(i)/m}). In expectation, this seems to correspond to the upper bound quantity appeared in the submission (equation (2)). In particular, it is not surprising to see the first part of the new quantity, as the empirical estimator does not produce any positive probability estimates below 1/m. Finally, the submission is lack of comparison with prior works, in terms of both results and proof techniques. Several papers are cited, including the ones above, but there is little discussion.

Correctness: I have checked some of the proofs or their sketches, which seem to be correct. On the practical side, the paper's contribution is mainly theoretical and does not contain any experiments.

Clarity: The learning problem and algorithm are intrinsically simple, and the presentation is clear. In my opinion, the paper is well written.

Relation to Prior Work: The submission lists a dozen papers, but provides little discussion and introduces most prior works by a single sentence. It would be helpful to include detailed reviews on some key references, such as the 2015 "On learning distributions" paper, together with comparisons.

Reproducibility: Yes

Additional Feedback:


Review 4

Summary and Contributions: This paper studies the problem of estimating an arbitrary discrete distribution, potentially supported on an infinite set of points, based on a finite number of samples m. There is a lot of previous work on this and related problems from a variety of perspectives. Most relevantly, Berend and Kontorovich gave guarantees for this problem which make sense for all such distributions (as opposed to subsets like bounded entropy distributions). The main result of this paper gives a guarantee like Berend and Kontorovich but with a bound computable just from the observed data. More precisely, they show that returning the empirical distribution has an error controlled by the half-norm of the empirical distribution and explain a minimax-type sense in which this is optimal. The proof of the upper bound more or less consists of two claims: (1) the risk is controlled by the empirical rademacher complexity of a related class (a standard fact) and (2) estimating the empirical rademacher complexity using basic facts about random walks.

Strengths: It is nice that the authors develop a simple bound which is easy to estimate empirically, and which has a simple and conceptually clean proof. The result implies a guarantee similar to the Berend and Kontorovich work (Theorem 2.3).

Weaknesses: The techniques used in the proof themselves are pretty standard. and the idea of using the half-norm itself has already appeared in the Berend and Kontorovich work.

Correctness: All of the proofs I read were clear and correct.

Clarity: Overall the paper is pretty readable.

Relation to Prior Work: The previous work is cited and discussed at a reasonable level of detail I feel, including explanation of where arguments in the proofs can be found.

Reproducibility: Yes

Additional Feedback: - The discussion of "instance optimality" is a bit confusing, I didn't really see how this is analogous to the Valiant and Valiant result. It seems more similar to a standard minimax result. - The notation o_p(1) used in Corollary A.1 is not defined as far as I could see. - (13) is missing an expectation ---- post-rebuttal: my opinion is unchanged

[Author Response · NeurIPS 2020]

We thank the reviewers for their encouraging comments and constructive feedback.

*Reviewer comment:* The empirical distribution estimator is often suboptimal. *Author response:* We thank the reviewer for their comment. We stress that the references Orlitsky et al. and Valiant and Valiant all assume an upper bound on the support size, and all their bounds break down for general distribution supported on countably many bins. Removing this assumption while still recovering proper convergence rates is our primary contribution. The empirical distribution still achieves the accurate minimax rate up to a universal constant for the learning problem with respect to total variation. We leave the investigations on improving the constant in our bounds and devising estimators that would be be first-order optimal as an interesting open question.

*Reviewer comment:* The results are similar to those in some intermediary steps of the prior work. Example of "On learning distributions from their samples" by Kamath et al. 2015, with an $O(\sqrt{(d-1)/m})$ upper bound on page 20, and an $O(\sum_i \sqrt{\mu(i)/m})$ upper bound just before. *Author response:* Indeed, the Kamath et al. 2015 paper is quite relevant to ours (it is already discusssed in the manuscript but we will expand the discussion in the revision). We note that their work is concerned with the finite support setting. Regarding the upper bound of the form $O(\sum_i \sqrt{\mu(i)/m})$, our response is three-fold. (i) Bounds of this form are well-known, appearing, e.g., in the Berend-Kontorovich 2013 paper we cite. (ii) Such bounds cannot handle general discrete distributions, as the sum diverges e.g., for $\mu(i) \propto 1/i^2$. (iii) Such bounds are not fully empirical (or adaptive) in the sense that they depend on the unknown $\mu$. As discussed in our Introduction, the main motivation behind our paper was to address the limitations of (i,ii,iii).

*Reviewer comment:* The submission lacks comparison with prior works, in terms of both results and proof techniques. *Author response:* We will expand the discussion within the space constraints, and add more detail about some key references. We stress that in most cases the results are not directly comparable, in that prior work has focused on the finite support case, and all the bounds (e.g. 2015 "On learning distributions") depend on the support size.

*Reviewer comment:* The nature of the bound does not allow a user to obtain risk bounds a priori before seeing the samples. *Author response:* As the reviewer points out, such a priori bound is provably impossible to obtain; we therefore disagree with this being considered a weakness of the paper.

*Reviewer comment:* Combine with some underlying structure / shape-constrained technology, when this structure is still itself insufficient to obtain convergent rates. *Author response:* We thank reviewers #1 and #2 for this constructive comment. We agree that it will be interesting to investigate the influence on the complexity measure of shape-constraints such as log-concavity, monotonicity, or unimodality of the distribution. It is known that distribution learning and testing complexities can change drastically under proper structure assumptions, and we leave the question of this influence on the empirical complexity measure as an exciting research direction.

*Reviewer comment:* Why is the main result presented in terms of the 1/2-norm instead of the empirical Rademacher complexity? *Author response:* The first reason is that the empirical half-norm is computationally inexpensive, whereas the empirical Rademacher complexity is harder to compute. The half-norm is also conceptually simpler to visualize than the Rademacher complexity, and can be instructively compared to Valiant & Valiant's two-third-norm for the identity testing problem.

*Reviewer comment:* Is there something that can be thought of as a crude analogue of the measure for continuous measures? *Author response:* We thank for the reviewer for this question. Investigating extensions of this methodology to continuous measures (e.g., via kernel density estimates) is an active research direction of ours.

*Reviewer comment:* Minor typos and clarifications. *Author response:* We thank the reviewers for their careful reading. We will clarify L69 and add the missing expectation symbol at (13). Additionally, we remove our $o_p$ claim (convergence in probability) in the remark at L160 and now simply assert that

$$\frac{\|\widehat{\boldsymbol{\mu}}_m\|_{1/2}^{1/2}}{\sqrt{2\pi m}} - \frac{3}{2}\sqrt{\frac{1}{2\pi}}\frac{1}{m^{3/2}}\|\widehat{\boldsymbol{\mu}}_m^+\|_{-1/2}^{-1/2} \leq \hat{\mathfrak{R}}_m(\boldsymbol{X}) \leq \frac{\|\widehat{\boldsymbol{\mu}}_m\|_{1/2}^{1/2}}{\sqrt{2\pi m}} + \sqrt{\frac{1}{2\pi}}\frac{1}{m^{3/2}}\|\widehat{\boldsymbol{\mu}}_m^+\|_{-1/2}^{-1/2}.$$

*Reviewer comment:* The discussion of "instance optimality" is a bit confusing, I didn't really see how this is analogous to the Valiant and Valiant result. It seems more similar to a standard minimax result. *Author response:* Indeed, the notion is distinct from that of Valiant and Valiant; we will clarify in the revision. Our intent was to draw attention to the fact that our empirical bounds are, in a sense, the best possible for *any* distribution, but perhaps *instance optimality* is not the best term for this.



[Meta-Review · NeurIPS 2020]

The paper provides data-dependent guarantees for learning discrete distributions, possibly with infinite support. Some of the steps in the argument appear in previous works (in particular the presence if the 1/2 norm), but the results are novel and I think there are some other technical tidbits that may be of interest to the community. I recommend acceptance.